# Serodiagnosis and Bacterial Genome of *Helicobacter pylori* Infection

**DOI:** 10.3390/toxins13070467

**Published:** 2021-07-05

**Authors:** Aina Ichihara, Hinako Ojima, Kazuyoshi Gotoh, Osamu Matsushita, Susumu Take, Hiroyuki Okada, Akari Watanabe, Kenji Yokota

**Affiliations:** 1Department of Bacteriology, Academic Field of Health Science Okayama University, Okayama 700-8558, Japan; p3ts632e@s.okayama-u.ac.jp (A.I.); pky34nw8@s.okayama-u.ac.jp (H.O.); 2Department of Bacteriology, Academic Field of Medicine Dentistry and Pharmaceutical Sciences, Okayama University, Okayama 700-8558, Japan; phwc8sst@cc.okayama-u.ac.jp (K.G.); osamu@okayama-u.ac.jp (O.M.); 3Department of Gastroenterology and Hepatology, Kurashiki Central Hospital, Kurashiki 710-8602, Japan; takesusumu@hotmail.com; 4Department of Gastroenterology and Hepatology, Academic Field of Medicine Dentistry and Pharmaceutical Sciences, Okayama University, Okayama 700-8558, Japan; hiro@md.okayama-u.ac.jp; 5Department of Oral Health Care and Rehabilitation, Institute of Biomedical Sciences, Tokushima University Graduate School, Tokushima 770-8504, Japan; akarita-maple@hotmail.co.jp

**Keywords:** antibody, VacA, CagA, genome

## Abstract

The infection caused by *Helicobacter pylori* is associated with several diseases, including gastric cancer. Several methods for the diagnosis of *H. pylori* infection exist, including endoscopy, the urea breath test, and the fecal antigen test, which is the serum antibody titer test that is often used since it is a simple and highly sensitive test. In this context, this study aims to find the association between different antibody reactivities and the organization of bacterial genomes. Next-generation sequences were performed to determine the genome sequences of four strains of antigens with different reactivity. The search was performed on the common genes, with the homology analysis conducted using a genome ring and dot plot analysis. The two antigens of the highly reactive strains showed a high gene homology, and Western blots for CagA and VacA also showed high expression levels of proteins. In the poorly responsive antigen strains, it was found that the inversion occurred around the *vacA* gene in the genome. The structure of bacterial genomes might contribute to the poor reactivity exhibited by the antibodies of patients. In the future, an accurate serodiagnosis could be performed by using a strain with few gene mutations of the antigen used for the antibody titer test of *H. pylori*.

## 1. Introduction

*Helicobacter pylori* is a Gram-negative microaerobic spiral-shaped bacterium that inhabits the mucous membrane of the stomach. About 44.5% of the world’s population is infected with *H. pylori* [1], and the infection causes inflammation of the gastric mucosa, resulting in gastritis, peptic ulcer, gastric MALT lymphoma and other gastrointestinal diseases. In fact, the infection caused by *H. pylori* causes inflammation of the gastric mucosa for a long period of time, atrophy and intestinal metaplasia progression, which is considered to be one of the causes of gastric cancer [2,3,4,5]. The infection caused by *H. pylori* affects more than half of the adult population worldwide [6], accounting for about 75% of all gastric cancer cases [7]. However, it is well-known that *H. pylori* infection incidence depends on geographic and other factors such as age, and race and socio-economic factors [8]. Gastric cancer is one of the common causes of cancer-related death. In addition, gastric cancer and peptic ulcers together cause more than 1 million deaths worldwide annually, the H. pylori infection being an important health concern [9]. Gastric cancer has a high prevalence in East Asian countries, including in Japan [10]. The early detection of *H. pylori* and its eradication are important for the prevention of gastric cancer. 

Serological tests constitute the most widely available non-invasive tests for the diagnosis of *H. pylori* infections. These tests are rapid to perform and inexpensive, making them useful for the screening of the population and also to confirm the presence of *H. pylori* infection in the case of ambiguous results from hemorrhagic ulcers, antibiotics, and/or other diagnostic methods with hypo-secretory therapy [11]. Generally, patients infected with *H. pylori* present specific circulating antibodies (IgG, IgA, IgM), which can be detected by specific serological tests. Currently, there are several commercial tests that are being developed, primarily based on IgG detection.

The developed antibody-based tests use antigens from one region of the world, which may not work well when applied to patients in other regions of the world and therefore may require local validation [12,13]. In this context, the reactivity of *H. pylori*- patients’ positive and negative sera was investigated in our laboratory using eight strains, derived from Japanese patients with different genotypes of *H. pylori* VacA toxin, adhesion factor, and CagA carcinogenic protein, as antigens [14]. In this study, two strains with high antibody reactivity and two strains with low antibody reactivity were selected from these bacteria, and the differences in their genes were investigated. Moreover, we investigated whether there is a relationship between the difference in antigen strain genes and the quality of antigen–antibody reactivity by determining the whole-genome sequence by using the next-generation sequence, comparing and examining the commonalities and differences among the four strains. Finally, the antigen–antibody reaction and the structure of the bacterial genomes were examined.

## 2. Results

### 2.1. Antibody Reaction of H. pylori Strains in the Patients 

The Az values of strains #3, #5, #6 and #8 assessed by ROC analysis of each antigen were 0.978, 0.991, 0.795 and 0.845, respectively (Figure 1). Strains #3 and #5 were used for analysis as strains with good antibody reactivity, and #6 and #8 as strains with poor reactivity.

### 2.2. Characteristics of Four Genome 

Table 1 shows the characteristics of the four antigen strains used in this study. The total DNA length was 1,585,790 bp for #3, 1,608,283 bp for #5, 1,630,060 bp for #6 and 1,668,167 bp for #8. The GC content was 38.8% for #3 and #8 and 38.7% for #5 and #6. The number of coding sequences (CDS) was, respectively, 1518, 1565, 1557 and 1607. The number of rRNA genes was four in all the four strains, while the number of tRNA genes was 36 in all four strains.

### 2.3. Number and Name of Common Genes

A percent identity of 90.0% or more and an overlap ratio of 95.0% or more were found as the common genes. The number of common genes was 1148 for #3 and #5, 1144 for #3 and #6, 988 for #3 and #8, 1219 for #5 and #6, 1026 for #5 and 998 for #8, #6 and #8. No significant difference was observed among the four strains; however, it was found that the number of common genes with #8 tended to be small.

Six genes for *eptA, ddl, ruvC, allr, ybeY,* and *xseB*, were found to be genes that were common to #3 and #5 and had a low concordance rate in #6 and #8. In addition, two genes—namely *exbD* and *metN*—were found to be genes that were common to #6 and #8 and had a low match rate between #3 and #5. Table 2 lists the molecular weight (Mw) and the function of these common genes.

### 2.4. Genome Ring

Genome rings were created to visualize the homology among the four genomes (Figure 2). The similarity of the ORFs was expressed at similar positions between #3 and #5. In #5 and #6, there were some places where similar ORFs were expressed, but there were more different parts compared to #3. Strain #8 had fewer ORFs in common in comparison with other strains. In the GC content genomic ring, sites with low GC content are shown inside. The GC content of the four strains was 38.7% to 38.8%, but there were some sites in which the GC content was significantly low, which may be due to the insertion of a foreign gene. Additionally, it was found that the insertion position differs depending on the strains.

### 2.5. Dot Plot Analysis

Dot plot analysis is a graph for comparing the amino acid sequence of a protein and the base sequence of a nucleic acid with each other, allowing clarification of the homology. The vertical axis is from bottom to top, while the horizontal axis is from left to right, and plots are at the matching points. Thus, the homology can be intuitively confirmed because the lines appear where the same sequence exists [15].

Dot plot analysis of #3 and #5 showed a straight line rising to the right (Figure 3A). In #3 and #6, the first and last sequences of the gene descended to the right, and a straight line rising to the right appeared in the center. This indicates that an inversion occurred (Figure 3B). Although some straight lines appeared, it was found that the homology was low in #3 and #8 (Figure 3C).

### 2.6. CagA and VacA Gene

The most important pathogenic genes of *H. pylori*—namely *cagA* and *vacA*—were visualized by a gene sequence map to investigate whether mutations occurred around them. Thus, the site that encodes about 30 genes, including *cagA*, which is involved in the pathogenicity of *H. pylori,* is called the cag pathogenicity island (*cagPAI*) [16]. No major mutations were found in four strains of *cagPAI*. However, it was found that the position of *cagPAI* started at about 1 million bp in the #8 strain, while starting at about 600,000 bp in the other strains (Figure 4). From the gene sequence map of *vacA*, it was found that the genes in *vacA* and its surroundings were in the forward direction in #3 and #5, and in the opposite direction in #6 and #8. Moreover, it was also found that the position of *vacA* was at about 1 million for #3 and #5, while it was at about 600,000 for #6 and #8, the positions also being different (Figure 5).

### 2.7. CagA and VacA Protein Expression Level

The expression levels of CagA protein and VacA protein in each strain were studied using Western blotting. The average expression level of CagA protein in three experiments of #3, #5, #6, and #8 were 231,884 ± 14,351, 151,335 ± 35,096, 63,947 ± 23,843, and 14,382 ± 7016 ECL (mean ± SE), respectively. It was found that the CagA expression level was lower in #6 and #8 than in strains #3 and #5. The statistical analysis revealed that the expression level was significantly lower in #8 than in #3 and #5 (*p*-value < 0.05). In addition, the expression level was significantly lower in #6 when compared with #3 (*p*-value < 0.05) (Figure 6A).

The average expression level of the VacA protein of #3, #5, #6 and #8 were 7842 ± 2584, 6029 ± 1883, 3196 ± 904 ECL and 3660 ± 1492 ECL, respectively. Similar to the results of the CagA expression level, it was found that the expression level of VacA was lower in #6 and #8 than in #3 and #5. From the four strains investigated, the VacA mutant #6 had the lowest VacA expression level. The statistical analysis demonstrated that no significant difference was found among the strains (Figure 6B).

## 3. Discussion

Several immune-dominant proteins, such as CagA, VacA, UreA, Oip and GroEL, have been used as candidates for detecting the *H. pylori* infection. The *H. pylori* FliD protein, an essential element of *H. pylori* in the assembly of its functional flagella, is also recognized as a novel marker for the serological diagnosis of *H. pylori* infection [17]. The immunoassay tests that use the six highly immunogenic virulence factors (CagA, VacA, GroEL, gGT, HcpC, and UreA) of *H. pylori* already demonstrated a high sensitivity and special envoys have been reported [16,18]. In this experiment, other common genes were identified for antigens with a high antibody reactivity and common genes for antigens with a low antibody reactivity, but they were not the above genes. Therefore, we suggest that, in the future, the proteins of these genes could be used as antigen candidates.

From the results obtained in this study, and the map around the genes of *cagPAI* and *vacA*, the possibility of mutations, such as inversion among the four strains, was considered. A mutation map was then created to make the mutations visually easy to understand (Figure 7). The red arrow represents the gene sequence in the same direction, while the blue arrow represents the site where the sequence is reversed. The gene at the position of 10,000 to 120,000 bp in #3 is at the end of 1.5 million bp in #5 and, conversely, the last sequence of #3 is at the beginning of #5. Most of the arrangements were found to be the same, and three reversal positions were identified. Comparing #3 and #6, the first half of the sequence of #3 is reversed in the second half of the sequence in #6; the second half of the sequence including *vacA* in #3 is reversed in the first half of the sequence in #6. It turned out that *vacA* gene existed in the reversed sequence. The central sequence containing *cagA* was found to be mainly homologous. Comparing #3 and #8, it was found that there were many mutations compared to the previous strains, #3, #5 and #6. Moreover, it was found that the positions of *cagA* and *vacA* were also very different in #8. Therefore, the position of the pathogenic gene perhaps differs between the strains with a high antibody reactivity and the strains with a low antibody reactivity. This phenomenon may affect the protein expressions of CagA and VacA; this is considered to be related to the accuracy of the diagnosis.

Antibody tests that use antigens from one single region of the world may not be appropriate when applied to patients from other regions of the world and thus may require a local validation [12,13]. Therefore, we analyzed all four strains isolated from Japanese patients, the antibody reactivity in the Japanese patients being different. The accuracy of the serological test is strongly dependent on the antigen used in the commercial kit and the prevalence of the particular *H. pylori* strain used as the source of the antigen. Appropriate antigens that use local strains as antigen sources or pool antigens from different groups of strains should be used. Moreover, it is important to establish reliable cutoff values for serological tests that should be locally validated before the investigation of the populations [19,20]. Finally, our study suggests that it is important to not only to use local antigens, but also to examine the genetic background of the antigens for serodiagnosis.

## 4. Materials and Methods

### 4.1. Strains

Antigens extracted from four Japanese-derived *H. pylori* strains were introduced into the SL-WAKO™ (Fujifilm Wako pure chemical corporation, Osaka, Japan) measurement system. *H. pylori*-negative sera (36 samples) and *H. pylori*-positive sera (56 samples) were used, and receiver operating characteristic curve (ROC) analysis was performed. The collected patients’ serum was used with the approval of the Okayama University Ethics Committee. 

### 4.2. DNA Extraction

The four strains stored at −80 °C were thawed and applied onto BHI agar medium containing 7% horse defibrillation blood and were then cultured in a 10% CO_2_ incubator for 1 week. DNA was extracted from the bacterial solution according to the product protocol of the Wizard^®^ Genomic DNA Purification Kit (Promega, Madison, WI, USA). The colonies were scraped with a sterile cotton swab, while 1 mL of physiological saline was placed in a 1.5 mL Eppendolf tube to suspend the bacteria, the cells were centrifuged at 15,000 rpm for 2 min, and then the supernatant was discarded. DNA was extracted according to the instruction manual.

### 4.3. Whole Genome Sequence

After confirming that 100 ng/µL or more of DNA could be extracted with Nano Drop 1000 (Thermo Fisher Scientific, Waltham, MA, USA), the Oral Resident Microflora Analysis Center (Kagawa, Japan) performed the analysis of the whole-genome through the sequence with a next-generation sequencer using the MiSeq system (Illumina, San Diego, CA, USA).

### 4.4. Genome Analysis

The obtained data were annotated using DFAST (https://dfast.nig.ac.jp/dfc/ (last accessed on 1 May 2021)). Gene analysis was performed using the In Silico Molecular Cloning Genomics Edition (IMC GE) software from In Silico Biology Co., Ltd. (version 7.54, Yokohama, Japan).

### 4.5. VacA and CagA Protein Expression

The four strains were applied to a BHI agar medium containing 7% horse defibrillation blood and were then cultured in a 10% CO_2_ incubator for 1 week. The colonies were scraped off with a sterile cotton swab, and then 5 mL of physiological saline was placed in a 15 mL Falcon tube to suspend the bacteria; the bacterial solution was then adjusted to an OD_600_ = 1.0 (10^8^ CFU/mL). The bacterial solution (1 mL) was added to a 1.5 mL Eppendolf tube, centrifuged at 10,000 rpm for 5 min, and then the supernatant was discarded. Samples were added to 50 µL of distilled water. Then, 50 µL of sample buffer was added and heated at 100 °C for 5 min to prepare a sample.

### 4.6. Western Blot

A sample of 5 µL was added to each lane, and the electrophoresis was performed at 90 V. Then, the proteins in the gel were transferred to a polyvinylidene fluoride membrane (PVDF) at 100 mA for 4 h. The PVDF membrane after transference was blocked with skim milk overnight. After washing 3 times with a phosphate buffered saline (PBS), primary antibodies—CagA antibody (Santa Cruz Biotechnology Inc., Dallas, TX, USA) and VacA antibody (Santa Cruz Biotechnology Inc., Dallas, TX, USA)—were reacted with 10% skim milk added to PBS. After washing 3 times with PBS, 10% of each HRP-labeled sheep anti-mouse IgG antibody (Thermo Fisher) (for CagA measurement) and HRP-labeled anti-rabbit immunoglobulin antibody (Dako, Denmark) (for VacA measurement) were used as secondary antibodies. The solutions were diluted 1000-fold with PBS that also contained skim milk and were shaken for 1 h. After washing 3 times with PBS, the color was developed by Pierce^®^ Western Blotting Substrate (Thermo Fisher), and the band was detected by chemiluminescence using the Amersham Imager 600 (GE Healthcare Japan Co., Ltd., Hino, Tokyo, Japan).

### 4.7. Statistical Analysis

A statistical analysis of CagA and VacA protein expression levels using Western blotting was performed using the Student’s *t*-test. If the p-value was less than 0.05, it was considered to be statistically significant.

## Figures and Tables

**Figure 1 toxins-13-00467-f001:**
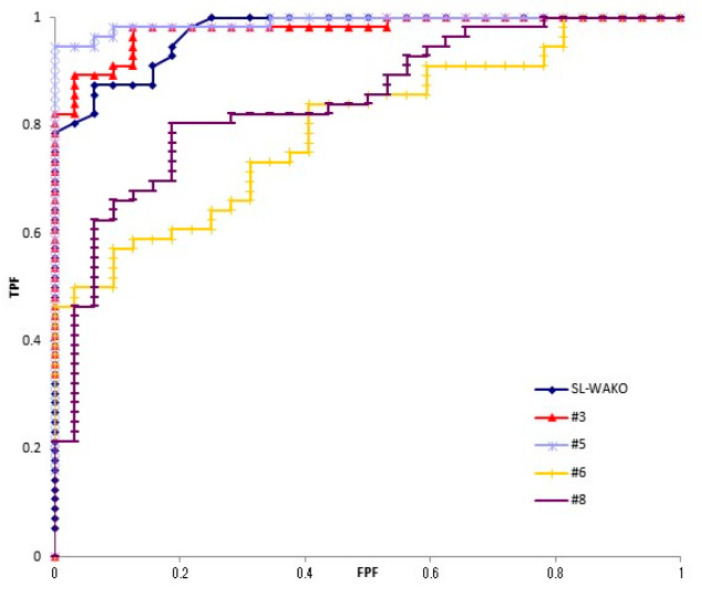
ROC curve of five strains. Four antigens from clinically isolated strains (#3, #5, #6 and #8) and kit original antigen (SL-Wako) were reacted with *H. pylori* negative and positive sera. ROC analysis was performed.

**Figure 2 toxins-13-00467-f002:**
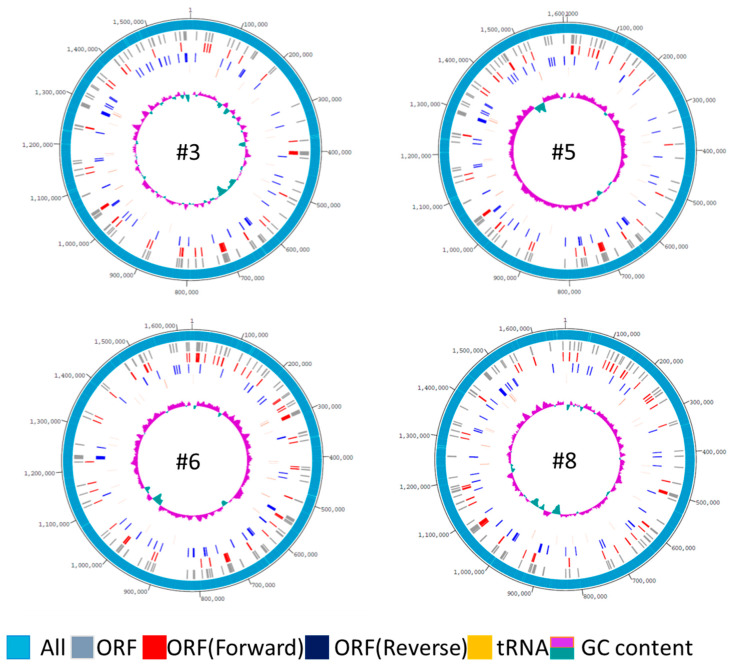
Genome rings. A genome ring was created to visualize the characteristics of each strain. From the outside, gray is the number of all ORFs, red is the forward ORF, blue is the reverse ORF, orange is the number of tRNAs, green is the number of rRNAs, and the innermost is the GC content.

**Figure 3 toxins-13-00467-f003:**
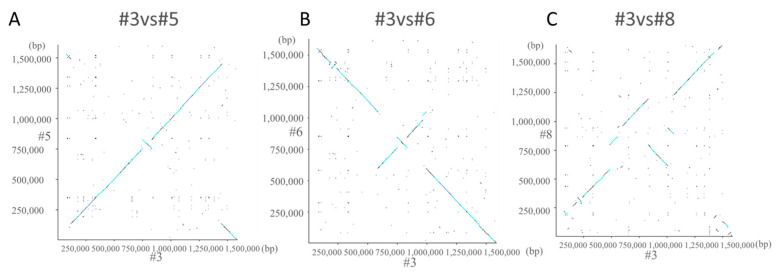
Dot plot analysis. Dot plot analysis was performed for #3 and #5 (**A**), #3 and #6 (**B**), and #3 and #8 (**C**). Dot plot analysis is a graph for comparing the amino acid sequence of a protein and the base sequence of a nucleic acid to clarify the homology. The vertical axis is arranged from bottom to top and the horizontal axis is arranged from left to right and is plotted at the matching points.

**Figure 4 toxins-13-00467-f004:**
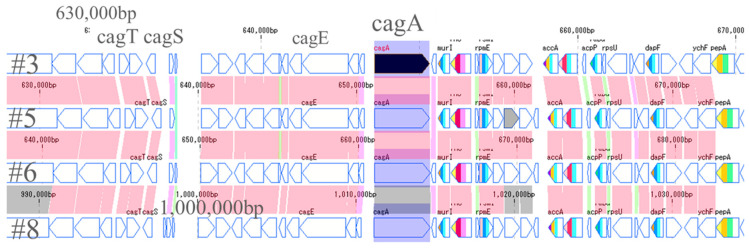
Comparison around *cagA* genes of 4 strains. The site that encodes about 30 genes including *cagA* is called cag pathogenicity island (CagPAI). No major mutation was found in 4 strains of CagPAI. However, it was found that the position of CagPAI started from about 1 million bp in the #8 strain and from about 600,000 bp in the other three strains.

**Figure 5 toxins-13-00467-f005:**
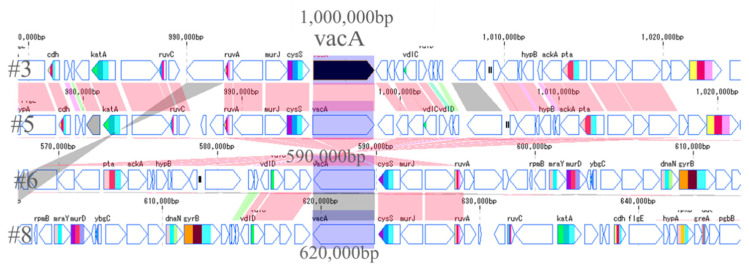
Comparison of *vacA* genes of 4 strains. The arrows of *vacA* and its surrounding genes are shown. The *vacA* genes of #3 and #5 were located at 1 million positions in the forward direction. In contrast, the *vacA* genes of #6 and #8 were present at 600,000 positions in reverse directions.

**Figure 6 toxins-13-00467-f006:**
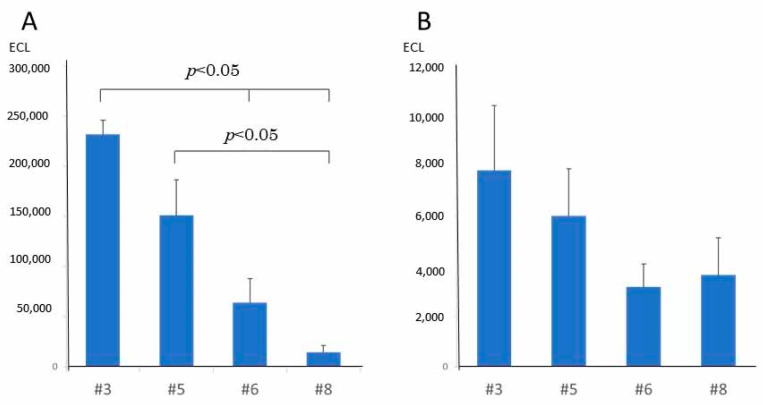
Protein productions of CagA (**A**) and VacA (**B**). The protein expression levels of CagA and VacA were examined using Western blotting (mean ± SE). The expression levels of CagA were significantly reduced at #6 and #8 compared to #3 and at #8 compared with #5. Although there was no significant difference in VacA expression level, there was, however, a tendency for the expression level to be low in #6 and #8.

**Figure 7 toxins-13-00467-f007:**
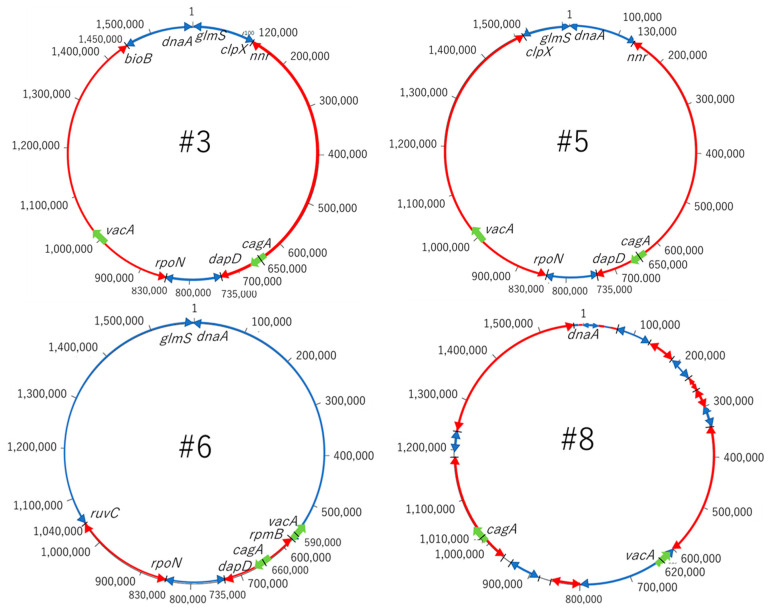
Chromosome structure. From the results of the dot plot analysis and the map around the genes of *cagPAI* and *vacA*, we considered the possibility that mutations, such as inversion, occurred among the 4 strains in the genome. Most of the sequences are the same for #3 and #5. It was found that the first half of the sequence of #3 was reversed in the second half of the sequence in #6, and the second half of the sequence including *vacA* in #6 was reversed in the first half of the sequence in #6. The middle sequence containing *cagA* should be approximately homologous. There are many mutations in #8 compared to #3, #5 and #6. The positions of *cagA* and *vacA* are also completely different in #8.

**Table 1 toxins-13-00467-t001:** Characteristics of strains.

Strain	Total Length (bp)	GC Content (%)	No. of CDSs	No. of rRNA	No. of tRNA
#3	1,585,790	38.8	1518	4	36
#5	1,608,283	38.7	1565	4	36
#6	1,630,060	38.7	1557	4	36
#8	1,668,167	38.8	1607	4	36

**Table 2 toxins-13-00467-t002:** Common genes.

Gene Name	Mw (kDa)	Function
Common between strain #3 and #5		
*eptA*	58.8	Resistance to antibiotics and metal ions
*ddl*	39.4	Metabolism of D-alanine, peptidoglycan Involved in biosynthesis
*ruvC*	17.4	Elimination of holiday structure
*alr*	41.8	Catalyzing the conversion of L-alanine and D-alanine
*ybeY*	16.1	Center of gene regulation via sRNA
*xseB*	7.6	Single-strand DNA endonuclease
Common between strain #6 and #8		
*exbD*	16.6	Involved in bacterial iron transport
*metN*	36.6	Involved in methionine uptake

## Data Availability

The complete genome sequence of #3, #5, #6, and #8 have been deposited in DDBJ.

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
