# Peer review of "Serodiagnosis and Bacterial Genome of Helicobacter pylori Infection"

_toxins, 2021, doi:10.3390/toxins13070467_

Round 1

Reviewer 1 Report

This manuscript describes interesting data on serodiagnosis and genomic characteristics of Hp in humans.

This manuscript is well-written and deserve minor modification before possible acceptance.

Global :

Prefer passive than active form

Genes have to be italicized.

Methods : 

Give versions for bioinformatic software. Moreover, they have to be appropriately referenced.

WGS part of the manuscript is inappropriately describe. What is the sequencing platform? Depth? Quality control process? Has the ORMACenter integrated in the authors' list? 

Do not use full capital letters for SANTA CRUZ BIOTECHNOLOGY.

All assays kits have to be correctly referenced (manufacturer, city, country)

Results :

The error bars correspond to IQR?SD?CI95? If CI95, it is pretty surprising that no association is statistically significant. If not CI95 please modify to consider CI95.

Author Response

Thank you very much for reviewing our paper.

A cording to your comments and questions, we have revised our paper.

  1. English sentences are natively checked by an English proofreading company. Please understand.
  2. The gene name has been changed to italics.
  3. WGS outsources to an external company for a fee. I know the system I used, so I added only that to the text. (page 259)
  4. Enter the manufacturer, city or state, and country name in the product name used.
  5. The Cag and VacA protein expression levels are shown as mean + SE. (page 175)

Reviewer 2 Report

Improving serological tests in diagnostics is very important for the earliest possible detection of the disease as well as for adequate treatment. There are a lot of grammatical and methodological errors in the article.

The same sentence appears in the introduction (line 52-53) and discussion (line 291-292). The discussion lacks comparisons with other research.

The usual abbreviation for bacteria was not used. We usually use H. pylori so it should change well. In materials and methods, it is not clear to me what you describe under 2.1. because you are not describing bacteria. That part needs to be rearranged and the graphic result put into materials and methods. Was the DNA extraction done according to the manufacturer's protocol? If it is, it should be stated and it does not need such a detailed description. In contrast, NGS needs to be described in more detail. Isn't it common to write gene names in italic? Where did the VacA mutant come from and how was it constructed? Why is the manufacturer of Santa Cruz Biotechnology written in capital letters? The text should also be proofread. 

Author Response

Thank you very much for reviewing our paper.

A cording to your comments and questions, we have revised our paper.

  1. We are not all native English speaker. So that, English sentences are natively checked by an English proofreading company. Please understand.
  2. I rewrote method of the DNA extraction.
  3. This is the first report of association with genome and antibody reaction in the patients. Although there are many reports of antibody reactions using individual proteins, there are few papers that examine the differences between strains and their genetic structure and diagnostic efficiency based on antibody titers, so the number of citations has decreased. I hope you understand.
  4. The gene name has been changed to italics.
  5. WGS outsources to an external company for a fee. I know the system they used, so I added only that to the text. (page 259)
  6. Enter the manufacturer, city or state, and country name in the product name used.

Reviewer 3 Report

NGS were performed to determined the genome sequences of four strains of different reactivity. The authors reported that mutations  are likely to affect the VacA and CagA production.

However, it will be interesting to compare these results with the characterization of the EPIYA motifs of the cagA gene and the vacAs1 and vacAm1  (or vacAs2 and vacAm2 allèles).

Conclusions cannot be drawn with only 4 strains.

Author Response

Thank you very much for reviewing our paper.

A cording to your comments and questions, we have answered below.

The structure, pathogenicity and antibody titer of CagA and VacA have been investigated using many strains. The four strains used in this experiment were selected from the following experiments.

The genetic diversity of Helicobacter pylori virulence genes is not associated with gastric atrophy progression. Masahide Kita, Kenji Yokota, et al. Acta Med Okayama 2013;67(2):93-98. (PubMed) In this paper, we used 58 strains.

We used selected 8 strains of them and examined the following.

strain 

Disease

vacA

iceA1

BabA2

cagPAI

cagE

cagA (EPIYA)

#1

DU

s1c/m1

+

+

+

A(B)D

#3

DU

s1c/m1

+

+

+

ABD

#5

DU

s1c/m1

+

+

+

ABD

#6

GU

s1a/m1

+

+

+

ABD

#8

DU

s2c/m2

+

+

+

AC

#11

GU

s1c/m1

-

+

+

AABDD

#14

DU

s1c/m1

+

+

+

AC

A Study to Determine the Optimum Antigens for the Serodiagnosis of Helicobacter pylori Infection in Japanese Patients and the Association with IgG Subclass and Gastric Cancer]. Kita M, et al.  Rinsho Byori. 2015 ;63(2):180-6.(PubMed in Japansese)

Conclusion of This paper is “The accuracy of serodiagnosis of H.  pylori  infection  may  increase  when  the  optimal  antigens  are  used,  and  measurement  IgG  subclass  may  provide additional prediction of gastric cancer.”

The 4 strains used in this experiment were originally selected from 58 strains that had undergone some genetic analysis, and we did not simply use 4 strains. I hope you will understand this.

Round 2

Reviewer 2 Report

The authors accepted all the comments and I have no further comments.